# The Vulnerability of International Floating Populations to Sexually Transmitted Infections: A Qualitative Study

**DOI:** 10.3390/healthcare11121744

**Published:** 2023-06-14

**Authors:** Jiahan Jiang, Yuyin Zhou, Junfang Xu, Zhaochen Wang

**Affiliations:** 1School of Public Health, Zhejiang University School of Medicine, Hangzhou 310058, China; eden56118@sina.com (J.J.); yuyinzhou0924@163.com (Y.Z.); 2Department of Pharmacy, Second Affiliated Hospital, Zhejiang University School of Medicine, Hangzhou 310009, China

**Keywords:** sexually transmitted infections, vulnerability, international floating population

## Abstract

With the rapid development of the global economy, along with globalisation, the health of international floating populations (especially their sexual health) has become a problem that cannot be ignored. This study explored the potential vulnerability of international floating populations to sexually transmitted infections (STIs) from the points of view of society, religion, culture, migration, community environment, and personal behaviours. In-depth exploratory interviews with 51 members of the international floating population living in China were conducted in June and July 2022. A qualitative thematic analysis methodology was used to analyse the content of these interviews. We found that a conservative culture orientated around religion leads to a lack of sex education, resulting in insufficient personal knowledge as well as a lack of the motivation and awareness required to encourage condom use during sexual contact. Additionally, both geographical isolation and reduced social supervision have expanded personal space, which has led to social isolation and marginalisation, in addition to challenges for coping with STI risk. These factors have increased the possibility of individuals engaging in risky behaviours.

## 1. Background

With the development of globalisation, population migrations are currently more frequent than ever before in human history. There are many reasons for migration, and though the factors that contribute to it are constantly changing, they are often related to the unequal distribution of resources, conflict, and changes in behaviour. Correspondingly, rapid cross-border migration and the resulting health problems—especially those related to infectious diseases—have become major policy issues in the 21st century [1,2]. As one group of infectious diseases, sexually transmitted infections (STIs) must be considered in the prevention and control of infectious diseases [3]. The Joint United Nations Programme on HIV/AIDS (UNAIDS) has attached great importance to transnational mobility as it relates to human immunodeficiency virus (HIV) as a key issue that must be addressed in order to end the global HIV epidemic by 2030 [4].

Moreover, given the emergence of an epidemic of sexually transmitted diseases (with AIDS being a key concern), governments have become worried that international floating populations may act as a medium for the spread of HIV, or that they themselves will become infected [5,6,7]. The term “international floating populations” refers to those who move to a foreign country but do not necessarily change their nationality or place of domicile, such as migrant workers and their relatives, as well as international students [8]. Given that most of the international floating populations residing far away from home are young (and, usually, healthy) people in a sexually active period of their lives, and that their contact with locals at their place of residence is inevitable, studies on STIs amongst the international floating populations have come to be of great importance.

From an international perspective, many studies have shown that international floating populations are more vulnerable than non-immigrants to STIs [9,10,11]. For instance, a Malaysian study indicated an increase of approximately 12 percentage points in the rate of HIV infection, in relation to migration [10]. Some quantitative studies have explored the impact of a single social factor, taking simple social, cultural, religious, and economic factors as their focus. For example, a Thai study analysed 2600 cross-border migrants aged 15–59, showing that mobility itself was not a major risk factor in HIV transmission [12]. Economic destabilisation under the conditions of mobility was found to be the main source of HIV transmission risk [12]. In addition, the international floating population has been reported to be more likely to engage in recreational drug use and high-risk sexual behaviours, with an increase in their number of sexual partners and the possibility of more risky sexual behaviours [13]. At the same time, due to language issues and other barriers, members of an international floating population often lack the motivation and skills to negotiate condom use during sexual contact [14].

However, there is a lack of comprehensive and in-depth research on the risk factors relating to the spread of STIs among international floating populations. Furthermore, evidence relating to the social and behavioural mechanisms behind STI transmission among the international floating population is still insufficient. Against this background, we have herein qualitatively explored the social and behavioural factors contributing to the risk of STI transmission among the international floating population.

According to incomplete statistics, as many as 560,000 foreigners undertake business activities in Yiwu every year, and 15,000 foreign businessmen from more than 100 countries and regions reside in Yiwu (excluding the duration of the COVID-19 epidemic). The structure of the immigrant population in Yiwu is complicated, with foreigners coming from more than 100 countries and regions, though many of them are Muslim [15]. Hangzhou, located in the eastern coastal region of China, is the capital of Zhejiang Province, where a high-tech service industry has rapidly developed in recent years, causing a large number of migrants to enter Hangzhou. Therefore, we chose Yiwu and Hangzhou as the settings for our research.

## 2. Methods

### 2.1. Study Design

To achieve a sufficiently deep understanding, we conducted in-depth, one-to-one, and semi-structured interviews with members of the international floating population, so as to explore the factors contributing to the risk of STI transmission, in addition to the constraints on high-risk behaviours in this cross-cultural context of mobility.

### 2.2. Participants and Setting

The participants of this study were members of the international floating population living in the cities Yiwu and Hangzhou, Zhejiang Province. As one of the largest commodity trading cities, about 80,000 international migrants live and conduct trade in Yiwu every year. It has become one of the most representative cities of transnational immigration, having the largest number of foreigners of any city in China. Hangzhou is the provincial capital of Zhejiang, and is located on the eastern coast of China, an area which has developed rapidly in recent years, in relation to the high-tech service industry. Correspondingly, a large foreign population has flooded into Hangzhou. All members of the international floating population residing in specific communities or hospitals during the period of our investigation were invited to participate in our study. The inclusion criteria were being over 18 years old and having the ability to communicate in either English or Chinese.

The international floating population of Yiwu mainly live in four communities: the Ci Lin community, Ji Ming Shan community, Si Ji community, and the Jin Cheng Gao Er Fu community. Most of them work in Yiwu International Trade City. Most of them have visited Fourth Affiliated Hospital of Zhejiang University for health care and medical treatment. Therefore, this study selected one of the four communities (i.e., Ji Ming Shan community), in addition to the Yiwu International Trade City and Fourth Affiliated Hospital of Zhejiang University, as the sources of interview participants. All members of the international floating population residing in the communities or visiting the hospital during the period of our investigation were invited to participate in our study. The interview was conducted in a separate room or space in Ji Ming Shan community, the Yiwu International Trade City, and Fourth Affiliated Hospital of Zhejiang University.

### 2.3. Interview Procedure

Semistructured interviews were conducted from 29 June to 4 July 2022 using a topic guide [5,7,8,10], but we also allowed the direction and content of each interview to be determined by each participant. The interview guide was designed by the principal researcher of this paper, using previous studies as a basis. It contained 10 questions, addressing the influence of social, cultural, religious, educational, migration-related, and community environmental factors on personal sexual behaviours, both before and after coming to China. The interviews were conducted in the specific hospitals or areas where the participants were residing. In total, 4 interviewers (including a leading interviewer) were chosen to conduct the interviews. The leading interviewers were experts on risky sexual behaviours and HIV, and conducted the interview in either English or Chinese, depending on the interviewee’s chosen language. At the start of the session, the leading interviewer explained the purpose of the interview, the terms of confidentiality, the format of the interview, and its expected length. All participants consented to the audio of their interview being recorded, and to the researcher taking brief notes. The duration of each interview varied from 20 min to one hour. Each interviewee was provided a portable fan as a compensation for their time.

### 2.4. Analysis

The qualitative thematic analysis methodology was used when analysing the interview content. Data analysis was undertaken following the procedures of thematic analysis, and a framework for assessing the vulnerability of the international floating population to sexually transmitted infections was built. The descriptive method was used to analyse the basic characteristics of participants using SPSS 23.0 software, with the provision of percentage and mean values.

### 2.5. Quality Control

In order to reduce non-cooperation in the interview process, the 4 interviewers were chosen carefully, ensuring they were proficient in English and Chinese, and equipped with knowledge regarding STIs. Moreover, they were trained before the interview on issues, such as the basic methods of interviewing, skills relevant to interviewing about sensitive issues, and an interview quality control. In addition, a set of online or face-to-face pre-interviews were conducted with 5 international floating populations in Guangzhou and Hangzhou, to ensure that the interviewees could understand the questions, assess their acceptance of different expressions of the questions, and ensure that the whole process was comfortable.

## 3. Results

### 3.1. Participants

A total of 51 international migrants residing in China participated in the interview (Table 1). Their ages ranged from 16 to 60 years old, with a mean of 36.8 and a median of 35. Their average time of residence in China was 10.4 years, ranging from 1 month to 29 years. In terms of participant’s highest qualifications, most participants held bachelor’s degrees or above (74.51%), followed by senior high school diplomas (19.61%), and below a junior high school qualification (3.92%). According to the National Bureau of Statistics of China, the average disposable income of urban workers for the year 2022 was about 4107 yuan. Among the interviewees, monthly income ranged from CNY 5000–6000 (four interviewees, 7.84%) to CNY 400,000 (one individual); 24 (47.06%) had no stable income. In terms of their Chinese proficiency, thirteen (27.8%) participants could undertake routine communication in Chinese, while four (8.33%) could hardly use Chinese. Most of the participants identified with Islam (68.63%), while ten (19.61%) followed Christianity, two (3.92%) followed Hinduism, and the rest professed no religion (7.84%). In total, eight (15.69%) interviewees were from Yemen, eight (15.69%) were from Iraq, two (3.92%) were from Egypt, two (3.92%) were from Libya, two (3.92%) were from Tanzania, two (3.92%) were from England, three (5.88%) were from Pakistan, three (5.88%) were from Russia, three (5.88%) were from Columbia, three (5.88%) were from Syria, and fifteen (29.41%) were from other countries.

### 3.2. Vulnerability before Migration

#### 3.2.1. Religious Belief

It was found that the participants’ religious beliefs may have had an impact on their level of sexual education. Via the interview, we found that all participants who misunderstood or had partial knowledge about sexually transmitted infections were Muslims and believed in Islam. A considerable portion (80.00%) of the Muslim participants were very resistant to discussing sexual topics when interviewed. A participant from Mali said: “No, I’m a Muslim. I don’t know [about sexual behaviour]”. Another participant from Pakistan said: “My religion tells me that I can’t do this [sex education for the next generation]. I’m a Muslim and no one tells me to do this”. The sexual culture of Islam is conservative, and Muslims adopt a serious attitude towards sexual issues [16]. Nearly half (42.86%) of the Muslims interviewed revealed that their parents and schools did not provide sexual health education, or at least did not provide sufficient and comprehensive education on these issues. Some (5.71%) of the Muslims interviewed were found to know nothing about premarital sexual behaviour, nor the issues relating to it (e.g., STIs, measures of contraception).

#### 3.2.2. Regional and Political Influence

In addition to religious belief, national context also has an impact on education. For example, among the interviewees, all immigrants from Syria said that “we had received sexual health education from our parents or schools, and think that sexual health education should be given to the next generation”. A Syrian immigrant said in their interview: “This [STI] has always been advertised in our country, telling us to be careful with these things. HIV is famous and everyone knows it”. Additionally, a study has shown that 44.7% of Syrian medical students mainly acquire knowledge on STIs through the media [17]. Effective public media usage has proven to be significantly associated with HIV knowledge and a lower stigmatisation of AIDS patients [18]. We can thus assume that Syria ensures the proliferation of STI-related knowledge through advertisements, which can help prevent the occurrence and reduce the prevalence of STIs. However, immigrants from Pakistan said that “we had not received relevant sexual health education, and are unwilling to provide it to the next generation. Pakistan was more traditional than open”.

#### 3.2.3. Openness of Sexual Culture

In the context of different sexual cultures, people’s attitudes towards sexual behaviour were also found to be different. In the interview, most Muslims (80.00%) declared that Islam prohibits premarital sex, and that their own sexual culture is more conservative than that of China: “We are Muslims. We can’t have premarital sex. It’s illegal”. The Muslims we interviewed were also resistant to discussing commercialised sexual behaviour and occasional sexual intercourse; a Muslim from Yemen said that “Other behaviours [referring to commercial sexual behaviour and temporary sexual partner behaviour] are few. People who have those behaviours are morally wrong”.

However, the cultures of specific countries also have imperceptible, far-reaching, and lasting impacts on sexual culture. Seven Islamic immigrants from various countries (e.g., Yemen, Tanzania, Morocco, and Colombia) claimed that premarital sex was allowed in their own country, and one of them stated a belief that China’s sexual culture was more open. Immigrants from Morocco also mentioned that “they do not mind talking about premarital sex, which is different from Muslim countries”. In comparison, Muslims from predominantly Muslim countries were found to be more conservative in terms of sex.

Muslim sexual culture emphasises sexual morality, prohibiting indulgence and promiscuity [16]. It advocates for and encourages the satisfaction of sexual desires through legal marriage. Accordingly, Islam has promulgated various regulations, such as the prohibition of private meetings between men and women who are not close relatives [16]. Many of the Muslims that were interviewed also cited the custom of separating men and women from childhood. The low incidence of sexually transmitted infections in Muslim countries may thus be closely related to their conservative sexual culture [16].

Among the interviewees, Christian immigrants espoused opposing attitudes toward sexual behaviour, compared to Muslims. They all stated a belief that premarital sex is very normal, and that commercialised sex and taking temporary sexual partners are common behaviours. An immigrant from Tanzania said, “Our country is open, […] we implemented polygamy. We often have two or three partners, which is different”. This open sexual culture should not be criticised, but it is worth noting that hidden health dangers accompany commercialised sexual behaviour and the taking of temporary sexual partners, which should not be ignored. If the relevant and necessary health detection and prevention measures are neglected, the risk of the spread of sexually transmitted infections increases.

#### 3.2.4. Lack of Sexual Health Education

Although almost all the participants (97.96%) showed a certain understanding of sexually transmitted infections, both misunderstandings and partial knowledge of STIs were still present. For example, STI can be transmitted through body fluids (such as the blood and breast milk). However, some participants wrongly believed that STIs could not be transmitted from mother to child, which is a common misunderstanding. Moreover, they were not clear about whether the virus could be transmitted through mosquitoes; in other words, they had only partial knowledge of STIs. In addition to the aforementioned beliefs, a Muslim from Egypt said that he had no idea what STIs, including HIV, were. This phenomenon of totally lacking an understanding of sexually transmitted infections is a manifestation of a lack of relevant education. Some previous studies have shown that a lack of STI-related knowledge might lead to insufficiencies in the awareness required to take preventive measures against STIs [19,20]. When people do not know about the routes of transmission of STIs or their symptoms, they cannot take effective preventive measures, nor can they seek timely medical treatment in the early stages of the disease in order to prevent its subsequent progression. The lack of adequate and comprehensive sexual health education was one of the main reasons for the shortage of such knowledge among participants.

#### 3.2.5. Age

In the interview, we found that the older the interviewees were, the weaker their awareness was of sexual health issues. This finding may be because sex education was relatively worse in earlier times. In 1993, the International Planned Parenthood Federation proposed in a research report that more extensive sexual health education should be carried out amongst adolescents. The United States and Japan both formulated comprehensive sex education manuals in the late 1970s, with more countries following suit shortly after [21]. Several sex education programs in the United States (such as the project entitled Taking Charge and Postponing Sexual Involvement) have also been launched since the 1980s [22]. It seems that prior to the end of the 1970s, global sexual health education had not received sufficient attention and promotion. Therefore, interviewees who were born in the period 1962–1979 may have received limited sexual health education.

### 3.3. Vulnerability after Migration

#### 3.3.1. More Open Sex Culture

A considerable number of participants (58.00%) mentioned that, compared with their original countries, China’s sexual culture is more open, with immigrants from Arabia and Libya stating that “Our country doesn’t like men and women to stay together”, and “We (men) can’t go out with women. We’ll get married if we choose someone”. These interviewees stated that in China, engaging in commercial sex and sexual behaviours with temporary partners is more acceptable and more likely to occur. A British immigrant said: “In China, commercial sex is better carried out (considering taxes and fees), and it is easier for people to have commercial sex in China than in Britain”. In a more open social environment, the informal mechanisms of social control (i.e., the social control implemented by conventional customs, ethics, public opinion, etc.) are weakened, acceptance of nonmarital sex is higher, and the barriers to communication and interaction between men and women disappear. People’s long-term experiences of being repressed and confined by a conservative national culture suddenly end, their personal space expands, and accordingly, individuals’ abilities to effectively manage sexual health-related risks also become challenged [23]. Further, the increased freedom to engage in commercial sex and sexual behaviour with temporary partners will make immigrants face a higher risk of contracting STIs [24].

#### 3.3.2. Family Member Supervision

All of the interviewees mentioned that the degree of freedom they experienced in all aspects of their lives increased after immigration. Nearly half (36.73%) of the immigrants reported that they had no relatives living in China, and 17 were living alone at the time of the study. When asked about changes in sexual behaviour before and after immigration, an international student from Thailand said: “I feel free in China. In my country, my parents controlled me. But here I can go out with my boyfriend and sleep with him, so it is more convenient. I can’t do it [having sex with my boyfriend] at home”. After emigration, due to the geographical isolation from their family, immigrants are extracted from the supervision of their parents (or spouses), and their personal space expands. The possibility of engaging in sexual behaviour (including commercial sex and sex with temporary partners) also increases, thus increasing the risk for STIs.

#### 3.3.3. Social Isolation and Marginalisation

In this study, the incomes of the respondents were generally in the upper–middle level, so it seems we cannot address the issue of marginalised social status caused by economic problems. Due to their geographical separation from their families and original social networks, while some respondents stated that the support they received from their families and friends had not changed, they did not deny that the number of contacts they had available had decreased. Nearly half (38.78%) of the immigrants admitted to feeling lonely and depressed because of language barriers and cultural differences; one immigrant from Syria complained that “I came to China to do business and have no real friends. When I first came to China, I didn’t find this. At first, I regarded others as friends, but (later) I found that they only wanted to do business with me”. A Polish immigrant also mentioned similar issues: “Generally speaking, it is difficult to make friends in Japan, South Korea, and China. People (in China) are very close to each other, but we are different. Everyone calls us foreigners”. It is difficult for migrants to fully integrate into a new society, resulting in a sense of separation and personal isolation that makes their vulnerability to STIs more complicated [23].

### 3.4. Suggestions for STI Prevention

Many interviewees mentioned that they would be willing to provide sexual health education for the next generation. An immigrant from Yemen pointed out that “this [sexual health education] is a cultural and educational issue”. They stated a belief that, among the most effective strategies for preventing STI transmission among international migrants, education is a top priority. A British businessman agreed, stating that “If international migrants and other people have not received education on this issue, they will really spread the disease freely. That is why it is very important to carry out comprehensive education in schools”. He added that during sex education in schools, teachers should not only teach the scientific aspects of sex, but also “interpersonal communication, safety, and consent between two people”.

## 4. Discussion

This study has explored the social and behavioural factors contributing to the risk of STI transmission within the international floating population (Figure 1). Unresolved language barriers at the personal level and the lack of related services at the structural level (such as insufficient medical services, the difficultly of access to information, and inadequate sexual health education) have exacerbated the complexity of STI vulnerability. At the social level, geographical separation from one’s original family and social network leads to the expansion of personal space; accordingly, the ability of migrants to cope with STI-related risks is also challenged. In addition, marginalised social status, a sense of isolation, and a sudden reduction in informal mechanisms of social control all further challenged the abilities of migrants to cope with the risks of STI. The spiritual and cultural aspects should also not be ignored. In line with earlier studies [23,25], we found that sexual culture had a determinative influence on STI risks. Before and after migration, religious beliefs and cultures continuously affected the formation of a sexual culture, the engagement with sexual health education, and the immigrants’ understanding of STIs. Other studies have shown that Muslims had less knowledge about STIs compared to other religious groups [26,27]. Our study confirmed that a conservative sexual culture, a lack of sex education and misunderstandings of STIs partly increased the risks of STI transmission.

In the context of religious beliefs and cultures, immigrants from conservative cultures tend not to receive adequate or comprehensive sexual health education during their childhood. Further, some older immigrants that we interviewed had not received sufficient sex education because people did not attach importance to this issue in the era during which they were educated, and they thus held negative and resistant attitudes towards sex education. The lack of sex education in schools may be contributing to the high rate of STIs in certain places [28,29,30]. Incomplete or lacking sexual health education on the part of migrants were found to have caused the development of misunderstandings and partial knowledge of sexually transmitted infections, and this lack of knowledge leads to directly influencing migrants’ ability to cope with or place limitation measures upon STI risk. Therefore, strengthening sexual health education in floating populations in Asia and elsewhere can help us prevent STI transmission [31]. In addition to teaching the basics, teachers should also teach students about the correct use of condoms and other prevention methods, as well as increase students’ knowledge about other ways STIs are spread (such as by sharing needles).

Although many of the interviewees thought that China’s medical services are relatively effective and need little improvement, they did admit that language barriers are a problem that cannot be ignored in relation to seeking medical treatment and access to information. The paucity of relevant services at the structural level, as well as limitations in the opportunities for patients to obtain information and medical care, hinder early intervention on the part of STI patients, thus further increasing the risk of STI transmission, which finding is consistent with previous research’s conclusions [32,33].

After migration, in the process of integrating into and adapting to Chinese society, the social marginalisation and isolation experienced by immigrants may aggravate the complexity of STI vulnerability, in addition to having a negative impact on the overall health of this population. Besides this effect, the expansion of personal space and reductions in informal social control mechanisms increased the possibility of immigrants having commercial sex or adopting temporary sexual partners, aggravating the risk of STI transmission; such findings were consistent with the results of prior studies [23,31,34,35,36]. We found that the relaxation in social control after migration was an important mediator between migration and AIDS risk-related behaviours, as has been shown in previous studies [36,37].

This study had some limitations. First of all, we only interviewed some of the immigrants living in Yiwu and Hangzhou, China, which may mean the results are not applicable to all immigrants in China. Second, we did not conduct a quantitative investigation, so we cannot verify the specific influence of each of the risk factors for STIs on international floating populations. Additionally, our study did not include all the important variables, such as English proficiency and the rate of refusing to be interviewed, which may also influence our explanation of these results. As such, other researchers should consider these factors when conducting future research. However, despite the above limitations, this study has preliminarily explored and addressed the complexity of vulnerability to STIs amongst the international floating population from various points of view. Furthermore, we have noticed that there are very few studies on the vulnerability of international floating populations to STIs in China. Our study may thus help to fill this gap. In addition, we conducted in-depth interviews, an approach relatively rare in previous studies. Finally, in the context of a large number of existent quantitative studies, our study was a qualitative study. It may thus provide a different perspective on the subject, in addition to adding research value.

## 5. Conclusions

This study provided a novel perspective on the complex vulnerability to STIs faced by the international floating population in China. The lack of sex education that results from religious and national cultural constraints, as well as the expansion of personal space, a sense of isolation, and a reduction in informal social control, all increased the challenges faced by migrants for coping with the risks of STI. Combined with the expressed wishes of some migrants, our findings emphasise that the international floating population should be considered as a group of concern for future public health intervention. Against the background of a lack of sexual health education (in combination with existing language barriers), the Chinese government may consider improving the publicity related to STIs in this community, so as to improve the risk awareness of immigrants and reduce the risks they face.

## Figures and Tables

**Figure 1 healthcare-11-01744-f001:**
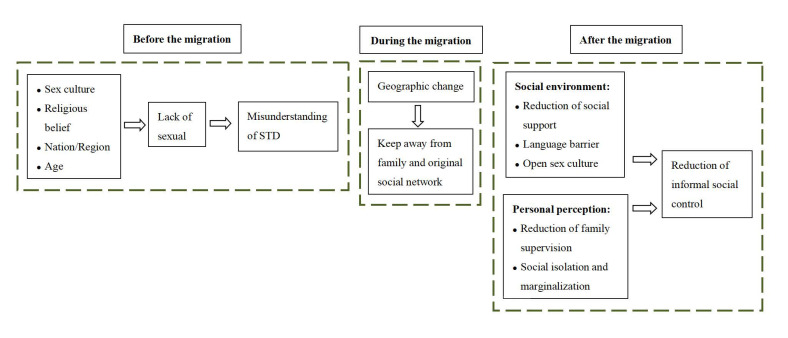
Framework for assessing the vulnerability of the international floating population to sexually transmitted infections.

**Table 1 healthcare-11-01744-t001:** Basic characteristics of interviewees.

Variables	Values
Total	51
Age (mean)	36.8 years
Age (median)	35 years
Age range	16~60 years old
Duration of residence in China (mean)	10.4 years
Duration of residence in China (median)	8 years
Residence time range	1 month~29 years
Education	
Bachelor’s degree or above	74.51%
Senior high school	19.61%
Below junior high school	3.92%
Personal monthly income	
CNY 5000–6000	4 (7.84%)
CNY 10,000–50,000	20 (39.22%)
CNY 155,000–400,000	3 (5.88%)
No stable income	24 (47.06%)
Chinese proficiency	
Speak Chinese fluently	15 (31.25%)
Conduct daily communication in Chinese	13 (27.08%)
Have some difficulties using Chinese for communication	16 (33.33%)
Hardly use Chinese	4 (8.33%)
Religion	
Christianity	10 (19.61%)
Islam	35 (68.63%)
Hinduism	2 (3.92%)
No religion	4 (7.84%)
Original country	
Egypt	2 (3.92%)
Pakistan	3 (5.88%)
Russia	3 (5.88%)
Columbia	3 (5.88%)
Libya	2 (3.92%)
Tanzania	2 (3.92%)
Syria	3 (5.88%)
Yemen	8 (15.69%)
Iraq	8 (15.69%)
England	2 (3.92%)
Other countries	15 (29.41%)

## Data Availability

All of the main data are included in the results. Additional materials with details may be obtained from the corresponding author.

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
