# Peer review of "The Vulnerability of International Floating Populations to Sexually Transmitted Infections: A Qualitative Study"

_healthcare, 2023, doi:10.3390/healthcare11121744_

Round 1

Reviewer 1 Report

The article is well-written and the topic is interesting. However, I think the authors should consider in the discussion not only eastern studies but also studies from Europe and other continents about the knowledge on STD and the most prevalent STD in the sexually active population. For example:

Ciccarese G, Drago F, Herzum A, Rebora A, Cogorno L, Zangrillo F, Parodi A. Knowledge of sexually transmitted infections and risky behaviors among undergraduate students in Tirana, Albania: comparison with Italian students. J Prev Med Hyg. 2020 Apr 2;61(1):E3-E5. doi: 10.15167/2421-4248/jpmh2020.61.1.1413. PMID: 32490261; PMCID: PMC7225650.

Vujcich D, Reid A, Brown G, Durham J, Guy R, Hartley L, Mao L, Mullens AB, Roberts M, Lobo R. HIV-Related Knowledge and Practices among Asian and African Migrants Living in Australia: Results from a Cross-Sectional Survey and Qualitative Study. Int J Environ Res Public Health. 2023 Feb 28;20(5):4347. doi: 10.3390/ijerph20054347. PMID: 36901357; PMCID: PMC10002009.

Santos-Hövener C, Marcus U, Koschollek C, Oudini H, Wiebe M, Ouedraogo OI, Thorlie A, Bremer V, Hamouda O, Dierks ML, An der Heiden M, Krause G. Determinants of HIV, viral hepatitis and STI prevention needs among African migrants in Germany; a cross-sectional survey on knowledge, attitudes, behaviors and practices. BMC Public Health. 2015 Aug 6;15:753. doi: 10.1186/s12889-015-2098-2. PMID: 26246382; PMCID: PMC4545823.

Ciccarese G, Herzum A, Rebora A, Drago F. Prevalence of genital, oral, and anal HPV infection among STI patients in Italy. J Med Virol. 2017 Jun;89(6):1121-1124. doi: 10.1002/jmv.24746. Epub 2016 Dec 23. PMID: 27935070.

Ciccarese G, Herzum A, Pastorino A, Dezzana M, Casazza S, Mavilia MG, Copello F, Parodi A, Drago F. Prevalence of genital HPV infection in STI and healthy populations and risk factors for viral persistence. Eur J Clin Microbiol Infect Dis. 2021 Apr;40(4):885-888. doi: 10.1007/s10096-020-04073-6. Epub 2020 Oct 16. PMID: 33067736.

Reviewer 2 Report

See attached.

Reviewer 3 Report

I have carefully read the manuscript by Jiang et al. and I find the study of the health needs of migrant people to be of great importance and interest, especially regarding sexual health, which is often ignored. In this specific work, rather than sexual health in general, the authors focused on analyzing only sexually transmitted infections (STIs).

The general recommendation is to review the entire manuscript linguistically. Some sentences, in my opinion, need to be rephrased (starting from the second phase of the introduction). Indeed, it is often hard to understand what the authors are communicating.

Some more specific comments:

Introduction

- When using acronyms for the first time (e.g., UNAIDS) it is necessary to write it in full;

- I recommend using the term "sexually transmitted infections" instead of "diseases" (throughout the manuscript);

- I think the background could be expanded (e.g., data on migration in the area where the study took place could help to understand the context)

Methods

- How were the participants selected? Was randomization conducted? How many agreed to participate out of the total number contacted?

Results

- Table 1 shows Chinese language skills. It would be interesting to show similar results for the English language, as it was also mentioned in the criteria for inclusion in the study.

Discussion

- Most of the discussion merely comments on the results of this manuscript as "in line with previous results." I recommend that the authors better emphasize this study's strengths. What does it add to the topic?

Ethics approval

- The approval number and date are missing.

Round 2

Reviewer 2 Report

healthcare-2171581-peer-review-v2

Review Report

General Comments

The English language in this revised manuscript has substantially improved. However, the Methods section needs revisions to give the reader the ability to replicate the process in their setting based on the lesson learned from this study. I am sending the Authors an article that clearly describes how 34 people were identified in South England and interviewed in their homes. The reader can clearly adapt this description to recruit individuals to participate in an interview research. I submit to the Authors to read the “Sampling and recruitment” as well as the “Data collection” sub-sections of this article and describe in like manner how the Authors identified potential participants, how they were initially contacted, how the 51 who participated were ultimately retained for the interviews, and where these interviews took place.

Specific Comments

Abstract

Line 3: Please make “explores” into “explored” so the report is about things that have already happened.

Second line from the bottom: “…and thus increases…”: what is the subject of the verb “to increase”? The reading is unclear here.

1. Background

Paragraph 2, lines 2 and 4 from the bottom of the paragraph: please make “population” plural, i.e., “populations”.

Paragraph 3, line 5 from the bottom of the paragraph (page 2): please insert “high-risk” before “sexual behaviors” so the phrase can read “…more likely to engage in drug abuse and high-risk sexual behaviors …”.

Paragraph 5, line 2 from the bottom of the paragraph (page 2): “setting” is a more specific term than “focus”, consistent with the Methods section.

2. Methods

2.2. Participants and setting

In my comments to the previous version of this manuscript, I stated that this section did not provide enough information to understand how participants were recruited in such a large city, only providing their age and their ability to communicate in English as inclusion criteria. I asked how were they identified to be invited to participate in the study, and where were the interviews conducted. In response, the Authors have added in this paragraph that “All members of the international floating population residing in specific communities or hospitals during the period of our investigation were invited to participate in our study”. This addition addressed none of the points raised, and added another question: did some members of the international floating population reside in hospitals? I refer the Authors to my general comment above and revise this section following the description in the article by Brien SB and colleagues.

2.3. Interview procedure

Were the interviews conducted during 2 days (in 29 June and 4 July 2022) or during more than 2 of the 6 calendar days between 29 June and 4 July 2022? In my comments to the previous version of this manuscript, I stated that this section did not describe where the interviews took place and whether participants were provided compensation for their time. In response, the Authors have added in this paragraph that “The interviews were conducted in the specific hospitals or areas where the participants were residing”. This addition does not bring specificity as to whether hospitalized respondents (if those who are said to reside in hospitals in section 2.2 above were in fact hospitalized) were interviewed at their hospital beds and where in the areas where the participants were residing the interviewers met them for the interviews.  Here again, I refer the Authors to my general comment above and revise this section following the description in the article by Brien SB and colleagues.

“The leading interviewers were experts on risky sexual behaviors and HIV, who…At the start of the interview, the interviewer explained…”. Was one interview conducted by a team of interviewers with a leading interviewer? Please describe what took place in enough detail to allow readers to understand the study procedures.

“The duration of each interview varied from 20 minutes to one hour”. In my comments to the previous version of this manuscript, I asked whether participants were provided compensation for their time. Please mention whether participants were given compensation for the 20 minutes to one hour of their time. If they were not compensated, please explain to international readers why that was the case so they can understand how participants were asked to give up to one hour of their time to participate in an interview that was not of interest to them.

2.5. Quality Control

Since none of the Authors conducted the interviews per the Authors’ contributions on page 9, how many “interviewers were chosen carefully” from the pool of experts referred to in Section 2.3, and how many conducted the interviews reported in this submission? I am asking these so readers can have a better understanding of the process by which 51 participants were interviewed within a 6-day time span from different residential areas including in hospitals, and each interview could take up to one hour.

3. Results

3.1. Participants

Line 2: Please replace “average” with “mean”, also in Table 1. And report also the median for both age and duration of residence in China.

Table 1, line 3 (page 4): “A mouth”: please replace “A” with “1”, a correct “mouth” into “month”. Also, most of the numbers associated with the percentages displayed in Table 1 are in the text. Please add these n’s also in the Table. I found the Table easier to look at with these n’s added, as I have added them by hand with my copy.

3.2.1. Religious belief

Line 3: “one-sided views about sexually transmitted infections”. In my comments to the previous version of this manuscript, I said under Section 3.2.4 that the reader could not understand what a one-sided interpretation of STDs meant because the concept had not been defined in the manuscript. In response to that comment from Section 3.2.4 below, the Authors have stated that “[some participants] were not clear about whether the virus could be transmitted through mosquitoes, which represent a one-sided interpretation; in other words, they had only partial knowledge of STIs”. I still did not understand whether lack of clarity and partial knowledge is what the Authors are saying is a one-sided view or interpretation. What is the reference that readers can check to assess the accuracy with which the concept of “one-sided interpretation” is used here? In other words, what do the Authors think will be lost here and in Section 3.2.4 if “one-sided interpretation” is removed from this manuscript?

Last line of the paragraph: “…in a pre-marital context.” The previous version had this as “before marriage”. Before marriage is much clearer than in a pre-marital context. For more clarity of expression, please consider returning “before marriage” here, as the previous version had it.

3.2.2. Regional and political influence

Last 4 lines in this section: “immigrants from Pakistan said…” and also “…They said Pakistan was…”. And similarly, in the 4th paragraph of Section 3.2.3 on page 6 it says “Immigrants from Tanzania said…”. There is no quote attributed to Pakistanis in this section but there is one attributed to immigrants from Tanzania, which raised the question of whether the 2 respondents gave that quote while being interviewed together. If the 2 were interviewed together, this should be described fully in the Methods that there were interviews that were taken by groups of respondents. A detailed description of the Methods will provide the context for understanding the Results and the Discussion. Let’s remember that the reader will need to fully understand how 51 individuals were interviewed within 6 days.

3.2.3. Openness of sexual culture

Paragraph 1, line 6: Please delete s in “Others behaviors” so the phrase reads “Other behaviors”.

Paragraph 2, lines 2-3: “Seven Islamic immigrants claimed that premarital sex was allowed in their own country,…” Were the seven from the same country? If yes, which country is it?

Paragraph 4, lines 4-5: “Immigrants from Tanzania said: ‘Our country is open, …we implemented polygamy. We often have two or three partners, which is different’”. Did both immigrants from Tanzania state this? If yes, were some participants interviewed as a group like in a focus group? (Please see related comment from Section 3.2.2 above)

3.2.4. The lack of sexual health education

In my comments under Section 3.2.1 above, I stated that I still did not understand whether lack of clarity and partial knowledge is what the Authors are saying is a one-sided view or interpretation. Because the Authors are using the concept of one-sided understanding as a “phenomenon” (line 9), “this one-sided phenomenon” should be referenced in this manuscript.

3.2.5. Age

Line 2: Please insert “be” between “may” and “because”.

3.3.1. More open sex culture

Second line from the bottom of the paragraph: “…faces immigrants with a higher risk of contracting sexually transmitted infections”. The previous version had this end of the paragraph more accurately with “…will make immigrants face a higher risk of sexually transmitted diseases”. For clarity of expression, please consider editing this portion of the text closer to what was in the previous version. Also, whether it is sexually transmitted infections or sexually transmitted diseases, at this point in the text, an abbreviated form (STIs or STDs) should be used instead.

3.3.2. Family member supervision

Second line from the bottom of the paragraph: Please consider moving “engaging in” between “of” and “sexual” so the whole statement reads: “The possibility of engaging in sexual behavior (including commercial sex and sex with temporary partners) is also increased, thus increasing the risk of STIs”.

4. Discussion

Paragraph 1, line 12: “…STI transmission”: here, “STI risk” is more precise; this study could not address transmission.

Paragraph 2, line 8: “one-sided beliefs”. Since the Authors have attached much importance to the concept of one-sided, at one moment even calling it a “phenomenon”, the concept must be referenced, as I indicated in my comments in earlier sections of the Results.

Paragraph 4, last line: Please consider a softer verb than “proven”.

Paragraph 5, line 4 (page 9) “…adoption risk sexual behavior”; there is something missing in this phrase. Please correct.

5. Conclusion

Line 1: Please consider a softer word than “evidence”.

Authors’ contributions: Please delete “all” so the last sentence reads: “All authors approved the…”

Figure 1 (page 5)

 There is no reference to Figure 1 made in the text. Please delete or make a reference to it in the manuscript. Did the Authors use the data obtained to construct a framework for assessing the vulnerability of international floating populations to STIs? They should state so, and explain the framework in the Discussion section.

Reviewer 3 Report

I thank the authors for addressing all my points. The manuscript in its current version is greatly improved. 

Author Response

I thank the reviewer again for all the helpful suggestions!